# p57$^{Kip2}$ is an essential regulator of vitamin D receptor-dependent mechanisms

**Katsuhiko Takahashi**[1,2☯], **Hitoshi Amano**[1,3,4☯]*, **Tomohiko Urano**[5,6☯], **Minqi Li**[7], **Meiko Oki**[3], **Kazuhiro Aoki**[3], **Norio Amizuka**[8], **Keiichi I. Nakayama**[9], **Keiko Nakayama**[10], **Nobuyuki Udagawa**[4], **Nobuaki Higashi**[1]

1 Department of Biochemistry, Hoshi University, Ebara, Shinagawa-ku, Tokyo, 2 Department of Anatomy, School of Medicine, Showa University Hatanodai, Shinagawa-ku, Tokyo, 3 Department of Basic Oral Health Engineering, Graduate School of Medical and Dental Sciences, Tokyo Medical and Dental University, Yushima, Bunkyo-ku, Tokyo, Japan, 4 Department of Biochemistry, Matsumoto Dental University, Shiojiri, Japan, 5 Department of Geriatric Medicine, Graduate School of Medicine, The University of Tokyo, Tokyo, Japan, 6 Department of Geriatric Medicine, School of Medicine, International University of Health and Welfare, Chiba, Japan, 7 Stomatology Department of Jining Medical University, Jining, and Department of Bone Metabolism, School of Stomatology Shandong University, Shandong Provincial Key Laboratory of Oral Tissue Regeneration, Jinan, China, 8 Developmental Biology and Hard Tissue, Graduate School of Dental Medicine and Faculty of Dental Medicine, Hokkaido University, Sapporo, Japan, 9 Department of Molecular and Cellular Biology, Medical Institute of Bioregulation, Kyushu University, Maidashi, Higashi-ku, Fukuoka, Japan, 10 Division of Cell Proliferation, ART, Graduate School of Medicine, Tohoku University, Sendai, Miyagi, Japan

☯ These authors contributed equally to this work.
* amano.bhoe@tmd.ac.jp

**Data Availability Statement:** All relevant data are within the manuscript and its Supporting information files.

**Funding:** This work was supported by JSPS KAKENHI Grant Numbers JP15K07950 (KT),

## Abstract

A cyclin-dependent kinase (CDK) inhibitor, p57$^{Kip2}$, is an important molecule involved in bone development; $p57^{Kip2}$-deficient ($p57$-/-) mice display neonatal lethality resulting from abnormal bone formation and cleft palate. The modulator 1α,25-dihydroxyvitamin D$_3$ (l,25-(OH)$_2$VD$_3$) has shown the potential to suppress the proliferation and induce the differentiation of normal and tumor cells. The current study assessed the role of p57$^{Kip2}$ in the 1,25-(OH)$_2$VD$_3$-regulated differentiation of osteoblasts because p57$^{Kip2}$ is associated with the vitamin D receptor (VDR). Additionally, 1,25-(OH)$_2$VD$_3$ treatment increased p57$^{KIP2}$ expression and induced the colocalization of p57$^{KIP2}$ with VDR in the osteoblast nucleus. Primary $p57$-/- osteoblasts exhibited higher proliferation rates with Cdk activation than $p57$+/+ cells. A lower level of nodule mineralization was observed in $p57$-/- osteoblasts than in $p57$+/+ cells. In $p57$+/+ osteoblasts, 1,25-(OH)$_2$VD$_3$ upregulated the $p57^{Kip2}$ and $opn$ mRNA expression levels, while the $opn$ expression levels were significantly decreased in $p57$-/- cells. The osteoclastogenesis assay performed using bone marrow cocultured with 1,25-(OH)$_2$VD$_3$-treated osteoblasts revealed a decreased efficiency of 1,25-(OH)$_2$VD$_3$-stimulated osteoclastogenesis in $p57$-/- cells. Based on these results, p57$^{Kip2}$ might function as a mediator of 1,25-(OH)$_2$VD$_3$ signaling, thereby enabling sufficient VDR activation for osteoblast maturation.

JP20K09474(HA), JP20K07789 (TU), JP19H01068
(KA), JP18H05215 (KIN), andJP19K07091 (NH).
The funders had no role in study design, data
collection and analysis, decision to publish, or
preparation of the manuscript.

**Competing interests:** NO authors have competing
interests.

## Introduction

Treatment with 1α,25-dihydroxyvitamin $D_3$ (l,25-$(OH)_2D_3$) suppresses the proliferation and induces the differentiation of osteoblastic cell lines. The induced growth inhibition was accompanied by a blockade of the transition from the G1 to S phase of the cell cycle [1]. We are interested in the regulatory role of l,25$(OH)_2D_3$ in cell growth and differentiation. The cyclin-dependent kinase inhibitor (CDKI) p57$^{Kip2}$ shares homology with the Cip/Kip family molecules p21$^{Cip1}$ and p27$^{Kip1}$ in the N-terminal domain (CDK inhibitory domain); additionally, CDKIs bind a variety of cyclin-CDK complexes and inhibit their kinase activities *in vitro* [2, 3]. Transfection of p57$^{Kip2}$ into Saos-2 osteosarcoma cells induced arrest at G1 phase through a mechanism that does not appear to require Rb or p53 [3]. Several reports have shown that SAOS2 cells were defective for p53 and Rb (Hinds et al. (1992), van der Heudel and Harlow [4, 5]). Decreased p57$^{Kip2}$ expression levels have been detected in several types of tumors [6–9]; consequently, p57$^{Kip2}$ might function as a tumor suppressor.

Mice deficient in the *p57$^{Kip2}$* gene exhibit defective endochondral bone formation. Most *p57$^{Kip2}$*-deficient mice die shortly after birth as a result of severe cleft palate. In studies using mice lacking Cip/Kip family CDKIs (p21$^{Cip1}$, p27$^{Kip1}$ and p57$^{Kip2}$), only *p57$^{Kip2}$*-deficient mice exhibited developmental abnormalities, including several defects attributed to abnormal bone formation [10–12]. Based on these results, p57$^{Kip2}$ might function as a cellular mediator of bone formation.

Bone resorption by osteoclasts is an important event in calcium homeostasis and bone metabolism. The maturation of osteoblasts involves mineralization and the upregulation of osteoclastogenic genes, and 1,25-$(OH)_2VD_3$, which is the most active metabolite of vitamin $D_3$, is a hormone required for osteoblast function [13–17]. This hormone exerts a positive effect on bone mass when administered *in vivo*. Osteoblasts express nuclear vitamin $D_3$ receptors (VDRs) and show increased expression of *osteopontin* (*opn*) and *rankl* after treatment with 1,25-$(OH)_2VD_3$. Osteoblasts are considered the major target of 1,25-$(OH)_2VD_3$ in bone [15,18]. Through interactions with nuclear VDRs, 1,25-$(OH)_2VD_3$ inhibits osteoblastic cell proliferation and upregulates the expression of VDR-dependent genes, including *opn* [19] and *rankl* [20], which increase the functional activity of mature osteoblasts to induce osteoclastogenesis.

The current study assessed the role of p57$^{Kip2}$ in the 1,25-$(OH)_2VD_3$-regulated differentiation of osteoblasts because p57$^{Kip2}$, but not other Cip/Kip molecules, was associated with VDR. In osteoblasts, VDR-dependent genes, including *opn* and *rankl*, were upregulated by VD$_3$-VDR, and *p57-/-* osteoblasts did not show upregulation of these genes. Because significantly higher levels of an osteoclastogenesis inhibitor, osteoprotegerin, were detected in *p57-/-* osteoblasts than *p57+/+* cells, p57$^{Kip2}$ may suppress *osteoprotegerin* (*opg*) expression in osteoblasts. Based on these results, we hypothesized that p57$^{Kip2}$ might be a mediator of 1,25-$(OH)_2VD_3$ signaling.

## Materials and methods

### Mice

Experimental animals carrying a targeted mutation in the p57$^{Kip2}$ loci were supplied by F. Hoffmann-La Roche, Ltd (Basel, Switzerland) [12]. The current study was approved by the Animal Use and Care Committee of Hoshi University. Mouse genotypes were determined by PCR. The p57$^{Kip2}$ locus is imprinted and expressed from the maternally derived allele. All experiments were conducted in accordance with the approval of the Hoshi University Animal Care and Use Committee (certificate number 20–070).

## Histology and immunohistochemistry

Dissected tissues were fixed with 4% paraformaldehyde in 0.1 M cacodylate buffer (pH 7.4). For the histological analysis, fixed samples were demineralized for 2 days in 10% formic acid, dehydrated with ascending concentrations of ethanol, cleared in xylene and embedded in paraffin wax. Embedded samples were sectioned at 5 μm; each section was retrieved from the water bath. For tartrate-resistant acid phosphatase (TRAP) enzyme histochemistry, the sections were deparaffinized, followed by immersion in 50 ml of an aqueous solution containing 5 mg of naphthol AS-BI phosphate (Sigma, St. Louis, MO), 25 mg of red violet LB salt (Sigma) and 100 mM L-(+) tartaric acid (0.76 g; Sigma) diluted in 0.1 M sodium acetate buffer (pH 5.4) for 15 min at 37˚C. For OPN immunohistochemistry, dewaxed sections were treated with 0.1% hydrogen peroxidase for 20 min to inhibit endogenous peroxidases; subsequently, dewaxed samples were pre-incubated with 1% bovine serum albumin in phosphate-buffered saline (pH 7.2; BSA-PBS) for 30 min at room temperature. Antisera against OPN (LSL Co., Tokyo, Japan) diluted 1:3000 were applied to the sections overnight at 4˚C, after which the specimens were incubated with horseradish peroxidase (HRP)-conjugated anti-rabbit IgG (Chemicon International Inc., Temecula, CA). Immunoreactions were visualized with diaminobenzidine as a substrate prior to observation under a light microscope. These sections were lightly counterstained with methyl green.

## Preparation of primary mouse osteoblasts

Primary cultures of calvarial osteoblasts were prepared using the sequential collagenase/dispase digestion method [21]. Briefly, calvaria were removed from newborn offspring derived from *p57+/-* males and *p57+/-* females; the neonates were denuded of soft tissue and digested with 1 mg/ml collagenase and 2 mg/ml dispase for 15 min at 37˚C in PBS with gentle agitation. The procedure was performed twice; cells from the second digestion were harvested and grown to confluence in α-MEM supplemented with 1% penicillin-streptomycin (Wako Pure Chemicals, Osaka, Japan) and 10% FCS.

## Immunoblot analysis

Cells were plated in 6-well plastic dishes and cultured with α-MEM containing 10% FCS, 5 mM β-glycerophosphate, 50 mg/ml ascorbic acid and antibiotics. The cells were treated with $1,25\text{-}(OH)_2VD_3$ (10 nM) for 24 h prior to collection for protein assays. Cells were rinsed twice with ice-cold phosphate-buffered saline and lysed with 180 μl of Nonidet P-40 lysis buffer [50 mM Tris-HCl (pH 7.4), 150 mM NaCl, 10 mM NaF, 5 mM EDTA, 5 mM EGTA, 2 mM sodium vanadate, 0.5% sodium deoxycholate, 1 mM dithiothreitol, 1 mM phenylmethylsulfonyl fluoride, 2 mg/ml aprotinin and 0.1% Nonidet P-40]; subsequently, the lysates were cleared by centrifugation at $15,000 \times g$ for 5 min at 4˚C. For immunoblot analyses, samples were separated on 9 or 12.5% SDS-PAGE gels. Immunoblotting was performed with an enhanced chemiluminescence detection system (GE Healthcare Life Sciences, Uppsala, Sweden). Anti-VDR and anti-*β*-Actin antibodies were purchased from Santa Cruz Biotechnology (Santa Cruz, CA). A rabbit antibody against p57<sup>Kip2</sup> was raised against the peptides corresponding to the C-terminus of the p57<sup>Kip2</sup> protein.

## Immunoprecipitation for the detection of the endogenous p57<sup>Kip2</sup>-VDR complex

Cells were plated in 60-mm plastic dishes and cultured with α-MEM containing 10% FCS, 5 mM β-glycerophosphate, 50 mg/ml ascorbic acid and antibiotics. The cells were treated with

1,25-$(OH)_2VD_3$ (10 nM) for 24 h prior to collection for protein assays. Cells were rinsed twice with ice-cold phosphate-buffered saline and lysed with 500 μl of RIPA buffer (1 mM sodium vanadate, 1% Nonidet P-40, 0.5% sodium deoxycholate, 0.1% sodium dodecyl sulfate, 1 mM phenylmethylsulfonyl fluoride and 2 mg/ml aprotinin in phosphate-buffered saline); subsequently, the whole lysates were cleared by centrifugation at $15,000 \times g$ for 5 min at 4˚C. For VDR immunoprecipitation, the lysates (200 μg protein) were mixed with anti-VDR antibodies (Santa Cruz, CA). The immunocomplexes were precipitated with Protein A/G-Sepharose beads (GE Healthcare Life Sciences); subsequently, the pellets were washed six times with ice-cold RIPA buffer. The precipitates were separated on 9% SDS-PAGE gels. Immunoblotting for the detection of both p57<sup>Kip2</sup> and VDR was performed with an enhanced chemiluminescence detection system (GE Healthcare Life Sciences, Uppsala, Sweden). Anti-p57<sup>Kip2</sup> and anti-VDR antibodies were purchased from Santa Cruz Biotechnology (Santa Cruz, CA).

**Immunofluorescence staining.** Primary mouse osteoblasts were treated with 10 nM 1,25 $(OH)$-$VD_3$ for 24 h. The cells were fixed with paraformaldehyde and incubated with the following primary antibodies: mouse anti-VDR (D-6: Santa Cruz Biotechnology) and rabbit anti-p57<sup>Kip2</sup> (EP2515Y, Abcam). After washing, the cells were incubated with the following secondary antibodies: Alexa Fluor 647-goat anti-mouse IgG and Alexa Fluor 488-goat anti-rabbit IgG (Jackson Immuno Reasearch). The cells were mounted with mounting medium containing DAPI to label the nuclei. Stained cells were detected by confocal microscopy (FV1200, Olympus); blue staining indicated DAPI-stained nuclei, red staining indicated VDR, and green staining indicated p57<sup>Kip2</sup>. The scale bar represents 100 μm.

## Nodule mineralization

Osteoblasts were plated in 12-well multiplates at a density of $2.5 \times 10^4$ cells/well and grown to confluence for 10 days. Media were then replaced with mineralizing media (α-MEM supplemented with antibiotics, 10% FCS, 10 μM β-glycerophosphate, 100 μg/ml ascorbic acid and 100 nM dexamethasone). Following an additional 3 weeks of culture [22], mineralization was detected with the von Kossa staining method. The von Kossa-stained area was measured with the application MetaMorph.

## Quantitative RT-PCR

The expression levels of the murine *p57<sup>Kip2</sup>* and *opn* mRNAs in mouse primary osteoblasts were evaluated with quantitative RT-PCR (qRT-PCR) using a Prism 7000 System (Applied Biosystems, Foster City, CA) and SYBR Green I fluorescence as previously described [23–25]. The expression levels of the human *p57<sup>Kip2</sup>* and *opn* mRNAs in SaOS2 cells were also assessed using qRT-PCR. The cDNA templates were synthesized from 1 μg of total RNA using the First Strand cDNA Synthesis Kit (GE Healthcare Life Sciences). The relative levels of the mouse and human *p57<sup>Kip2</sup>* and *opn* mRNAs, which were normalized to the reference gene *hypoxanthine-guanine phosphoribosyl transferase* (*hprt1*), were determined using the comparative Ct (cycles at threshold fluorescence) method. All experiments were independently repeated three times, i.e., each experiment was performed in triplicate. The sequences of the PCR primers are as follows: mouse *p57<sup>Kip2</sup>* (forward 5ʹ–AACCGCTGGGACTTCAACTTC-3ʹ, reverse 5ʹ–AGACTCGCTGTC CACCTCCAT-3ʹ), mouse *opn* (forward 5ʹ–CCCTCGATGTCATCCCTGTT-3ʹ, reverse 5ʹ–CTGCCCTTTCCGTTGTTGTC-3ʹ), mouse *hprt1* (forward 5ʹ–TGGGAGGCCATCACATTGT-3ʹ, reverse 5ʹ–AGCAGGTCAGCAAAGAACTTATAGC-3ʹ), mouse *rankl* (forward 5ʹ-CCAG CATCAAAATCCCAAGTTC-3ʹ, reverse 5ʹ–TGCCCGACCAGTTTTTCG-3ʹ), mouse *opg* (forward 5ʹ–GCCTGGGACCAAAGTGAATG-3ʹ, reverse 5ʹ–CTTGTGAGCTGTGTCTCCGTTT-3ʹ), human *p57<sup>KIP2</sup>* (forward 5ʹ–AGTCCCTCGACGGCCTCGAG-3ʹ, reverse 5ʹ–

CGGGACCGGGACACTAGGCA-3'), and human *opn* (forward 5'-ATGAGCATTCCGATGT GATTG-3', reverse 5'-TGTGGAATTCACGGCTGA-3').

## Constructs and transfection

Expression plasmids for amino terminally HA-tagged VDR (HA-VDR) were constructed by ligating the cDNA fragments into the FLAG-pcDNA3.1(-) vector. For transfection, the human osteosarcoma cell line SaOS2 and its tetracycline (tet)-off p57$^{Kip2}$ stable transfectant were used. Cells ($5 \times 10^5$) were grown in 100-mm culture dishes; Lipofectamine (Invitrogen) was employed according to the manufacturer's instructions. Cells were harvested or analyzed after 48 h. Immunoblotting and immunoprecipitation were performed as described previously [26]. For immunoprecipitation, the cell lysates were mixed with anti-FLAG M2 affinity gel (Sigma); FLAG-tagged proteins were eluted with the $3 \times$ FLAG peptide after washing. The eluate was subjected to immunodetection utilizing anti-FLAG, anti-VDR and anti-p57$^{Kip2}$ antibodies.

## Statistical analysis

All data were analyzed with Student's t-test and two-way ANOVA. Differences were considered statistically significant when $p < 0.01$.

## Results

### The VD$_3$-dependent interaction between VDR and p57$^{Kip2}$

We hypothesized that the 1,25-(OH)$_2$VD$_3$ (VD$_3$) activity in osteoblasts might depend on the p57$^{Kip2}$ protein. In the immunoprecipitation analysis of the primary mouse osteoblasts, p57$^{Kip2}$ was detected in the complex precipitated with anti-VDR antibodies in the VD$_3$-stimulated osteoblasts (Fig 1A). VD$_3$ stimuli also induced the colocalization of p57$^{Kip2}$ with VDR in the nuclei of the primary mouse osteoblasts (Fig 1B).

### p57$^{Kip2}$ enhances the transcriptional activities of VDR

Next, we used SaOS2 cells characterized by tet-off regulation of p57$^{Kip2}$ to assess the effects of p57$^{Kip2}$ on VDR activation. In humans and rodents, the expression of *opn* transcripts depends on the formation of the 1,25-(OH)$_2$VD$_3$-VDR complex at the VDR response element (VDRE) [27, 28]. In the absence of tet, the expression of both the *p57$^{Kip2}$* and *opn* mRNAs was upregulated (Fig 2A and 2B). In the presence of tet, *p57$^{Kip2}$* transcripts were also downregulated by the tet-off system (Fig 2A). Because *opn* transcripts were simultaneously decreased in the presence of tet, p57$^{Kip2}$ expression increased the levels of the *opn* mRNA (Fig 2B). We performed a luciferase assay employing a reporter plasmid with the *opn* 5' flanking region-luciferase cDNA to investigate the role of p57$^{Kip2}$ in the activation of the VDRE. Following cotransfection of the reporter plasmid and VDR expression plasmids into SaOS2 tet-off p57$^{Kip2}$-stable cells, luciferase activity increased by two-fold upon the addition of 1,25-(OH)$_2$VD$_3$. Our SaOS2 tet-off p57$^{Kip2}$-stable cells may have expressed p57$^{Kip2}$ at sufficient levels to activate the *opn* 5' flanking region when the cells were cultured in medium lacking tet. The p57$^{Kip2}$-on cells might have expressed p57$^{Kip2}$ at excessively high levels that were unable to increase the activity of the *opn* 5' flanking region through VDR expression, and the *opn* 5' flanking region was strongly activated by 1,25-(OH)$_2$VD$_3$ (Fig 2D). Under the tet-off condition (basal levels of p57$^{Kip2}$), VDR expression was sufficient for *opn* promoter activation by 1,25-(OH)$_2$VD$_3$; furthermore, the induction of both VDR and p57$^{Kip2}$ expression increased the activation of the promoter after the 1,25-(OH)$_2$VD$_3$ treatment (Fig 2C). Thus, p57$^{Kip2}$ activated the *opn* 5' region, and the coexistence of p57$^{Kip2}$ and VDR increased the 1,25-(OH)$_2$VD$_3$-induced activation of *opn*

**A**

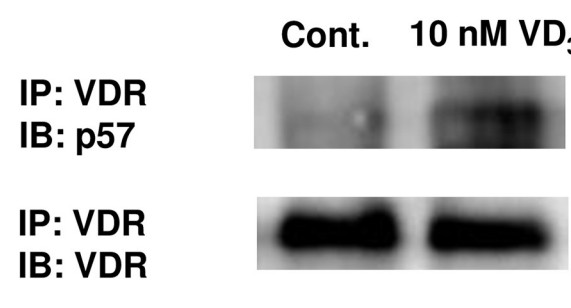

**B**

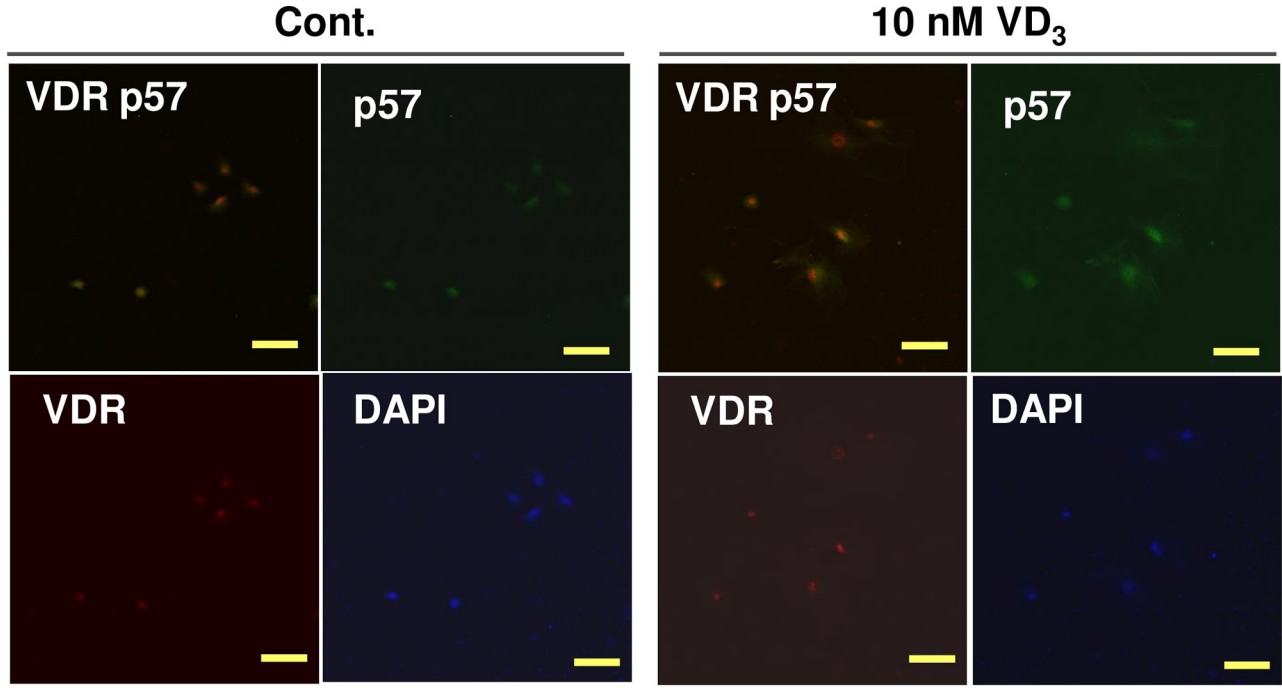

**Fig 1. The association of p57<sup>Kip2</sup> with VDR is dependent on 1,25-(OH)₂VD₃.** A. The association of p57<sup>Kip2</sup> with VDR is dependent on 1,25-(OH)₂VD₃ in primary p57+/+ osteoblasts. Top photo: immunoprecipitation (IP) was performed with an anti-VDR antibody, and immunoblotting (IB) was conducted with an anti-p57<sup>Kip2</sup> antibody. Bottom: immunoprecipitation and immunoblotting were performed with an anti-VDR antibody. B. Immunofluorescence staining of primary mouse osteoblasts that were treated with 10 nM 1,25-(OH)₂VD₃ for 24 h. Blue indicates DAPI-stained nuclei, red indicates VDR, and green indicates p57<sup>Kip2</sup>. The scale bar represents 100 μm.

expression. Osteogenic homeostasis mediated by 1,25-(OH)₂VD₃ might be sufficient in osteoblasts expressing both p57<sup>Kip2</sup> and VDR.

## Effects of the ablation of p57<sup>Kip2</sup> in primary cultured osteoblasts

We prepared primary calvarial osteoblasts harvested from neonates to examine the role of p57<sup>Kip2</sup> in osteoblast maturation in detail. The primary *p57-/-* osteoblasts displayed a higher

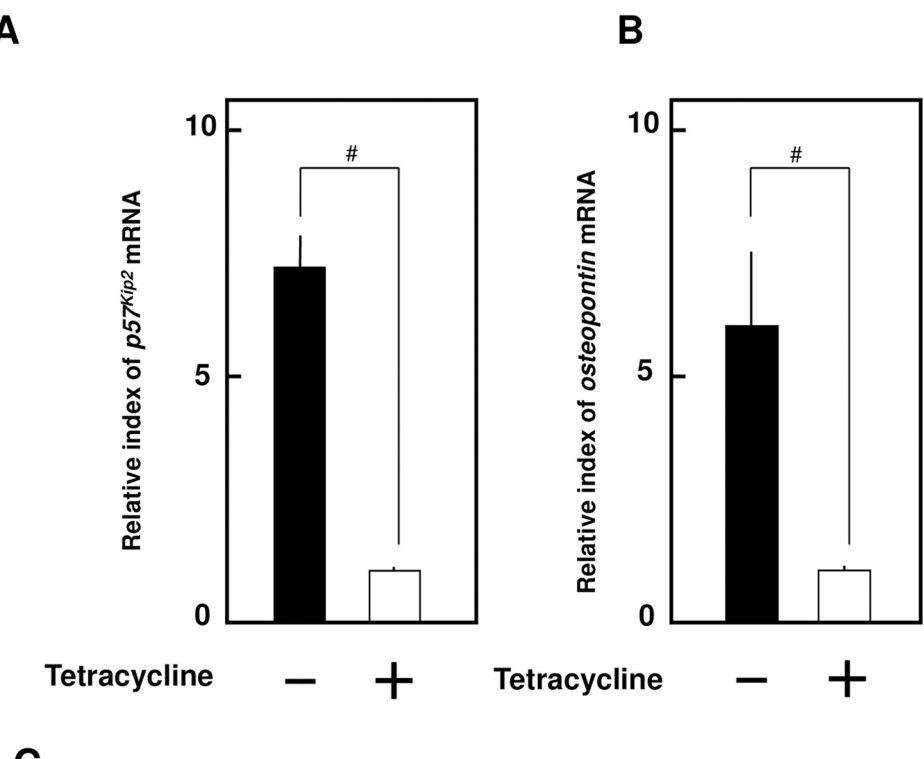

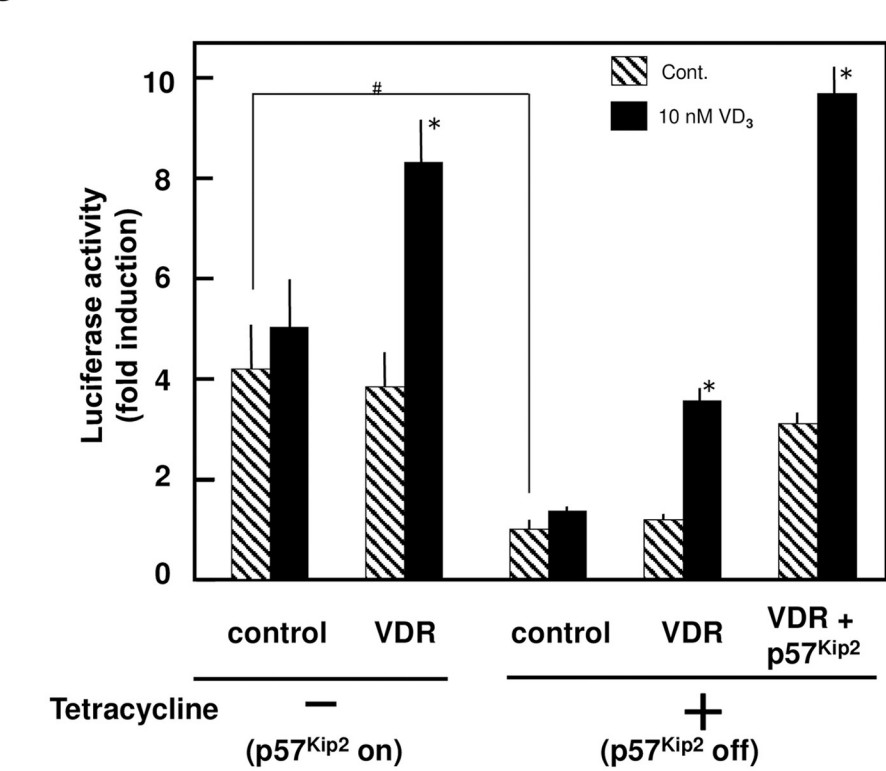

**Fig 2. p57$^{Kip2}$ Upregulates the VD$_3$-dependent expression of *opn* transcripts.** Expression levels of both *p57$^{Kip2}$* (A) and *opn* (B) depended on the presence of tetracycline in the SaOS2 tet-off p57$^{Kip2}$ transfectant. An asterisk indicates statistical significance: #, *p* < 0.01. (C) The activity of the *opn* promoter was estimated in SaOS2 tet-off p57$^{Kip2}$-transfected cells. In the SaOS2 tet-off p57$^{Kip2}$ transfectant, p57$^{Kip2}$ expression (- tetracycline) upregulated *opn* promoter activity independently of 1,25-(OH)$_2$VD$_3$. Under equal transfection conditions, an asterisk indicates a

statistically significant difference between 0.1% ethanol and 1,25-(OH)$_2$VD$_3$: *, $p < 0.01$. Under equal culture conditions, a # symbol indicates a statistically significant difference among transfection conditions: #, $p < 0.01$. The open columns represent the vehicle (0.1% ethanol: EtOH), whereas the closed columns represent 10 nM 1,25-(OH)$_2$VD$_3$.

proliferation rate than wild-type (*p57+/+*) cells (S Fig 1 in S1 File). The continuous induction of differentiation in the confluent osteoblasts was observed in an *in vitro* culture for three weeks. After extended culture, *p57-/-* osteoblasts exhibited lower levels of mineralization than *p57+/+* cells. The levels of p57$^{Kip2}$ transcripts and proteins in cultured primary mouse osteoblasts were assessed after treatment with 1,25-(OH)$_2$VD$_3$. The qRT-PCR analysis revealed the upregulation of the *p57$^{Kip2}$* transcript in 1,25-(OH)$_2$VD$_3$-treated mouse osteoblasts (Fig 3A). Concomitant with the increase in the levels of *p57$^{Kip2}$* transcripts, substantially increased levels of the p57$^{Kip2}$ protein were detected using immunoblotting (Fig 3B). These findings were consistent with a previous report noting the stabilization of p57$^{Kip2}$ protein in rat osteoblasts in the presence of 1,25-(OH)$_2$VD$_3$ [29]. Subsequently, we hypothesized that 1,25-(OH)$_2$VD$_3$ activity in osteoblasts might depend on the levels of the p57$^{Kip2}$ protein. An analysis of *opn* mRNA levels in *p57+/+* osteoblasts in the early phase of mineralization revealed an upregulation of *opn* transcripts, which was increased two-fold by 1,25-(OH)$_2$VD$_3$. In *p57-/-* cells, the induction of differentiation did not lead to increased *opn* expression; however, the 1,25-(OH)$_2$VD$_3$ treatment induced *opn* expression (Fig 3C). Thus, the expression of *opn* transcripts in osteoblasts might depend on VDR and p57$^{Kip2}$. Nodule mineralization was observed during the extended culture of both *p57-/-* and *p57+/+* cells; furthermore, the mineralized nodules in *p57-/-* cells were significantly smaller than those in *p57+/+* cells (Fig 3D). Additionally, the area of the mineralized nodules was reduced by approximately 30% (Fig 3D and 3E). Primary *p57+/+* osteoblasts expressed *opn* mRNA at higher levels than *p57-/-* cells, and the differences were more obvious in confluent osteoblasts stimulated with mineralization medium (Fig 3F). Immunohistochemistry findings demonstrated lower levels of Opn-expressing osteoblasts in the bone medulla of *p57-/-* neonates than in *p57+/+* neonates (S Fig 2 in S1 File). Based on these results, we hypothesized that p57$^{Kip2}$ might be necessary to ensure the sufficient bioactivity of 1,25-(OH)$_2$VD$_3$ during osteoblastic maturation.

## p57-/- osteoblasts exhibit defects in osteoclastogenesis

We hypothesized that p57$^{Kip2}$ might play a role in the 1,25-(OH)$_2$VD$_3$-induced osteoclastogenesis of osteoblasts. We cocultured 1,25-(OH)$_2$VD$_3$-treated osteoblasts with bone marrow cells to identify the roles of p57$^{Kip2}$ in the osteoclastogenic activity of osteoblasts. Notably, 1,25-(OH)$_2$VD$_3$ stimulated osteoclastogenesis in *p57+/+* primary osteoblasts more effectively than in *p57-/-* osteoblasts (S Fig 3 in S1 File). Based on these findings, p57$^{Kip2}$ might be necessary to maintain proper mineralization and p57$^{Kip2}$ might promote VD$_3$ signaling. Receptor activator of NF-kappa B ligand (Rankl) is a membrane-bound signal transducer responsible for the differentiation and maintenance of osteoclasts. In *p57+/+* osteoblasts, *rankl* transcripts were upregulated 4.7-fold after 1,25-(OH)$_2$VD$_3$-treatment (72 h) (Fig 4A). Opg, also known as an osteoclastogenesis inhibitory factor, functions as a decoy receptor for Rank to obstruct Rankl-Rank signaling and inhibit osteoclastogenesis. The expression of *opg* transcripts was significantly increased in *p57-/-* osteoblasts (Fig 4B). *Rankl* expression might depend on p57$^{Kip2}$ to some extent, and *opg* expression might be suppressed by p57$^{Kip2}$. Thus, the defects in *p57-/-* osteoblasts might result from disturbances in *rankl* and *opg* expression levels. Additionally, 1,25-(OH)$_2$VD$_3$ upregulated *rankl* expression by 4.7-fold in *p57+/+* cells and 1.4-fold in *p57-/-*

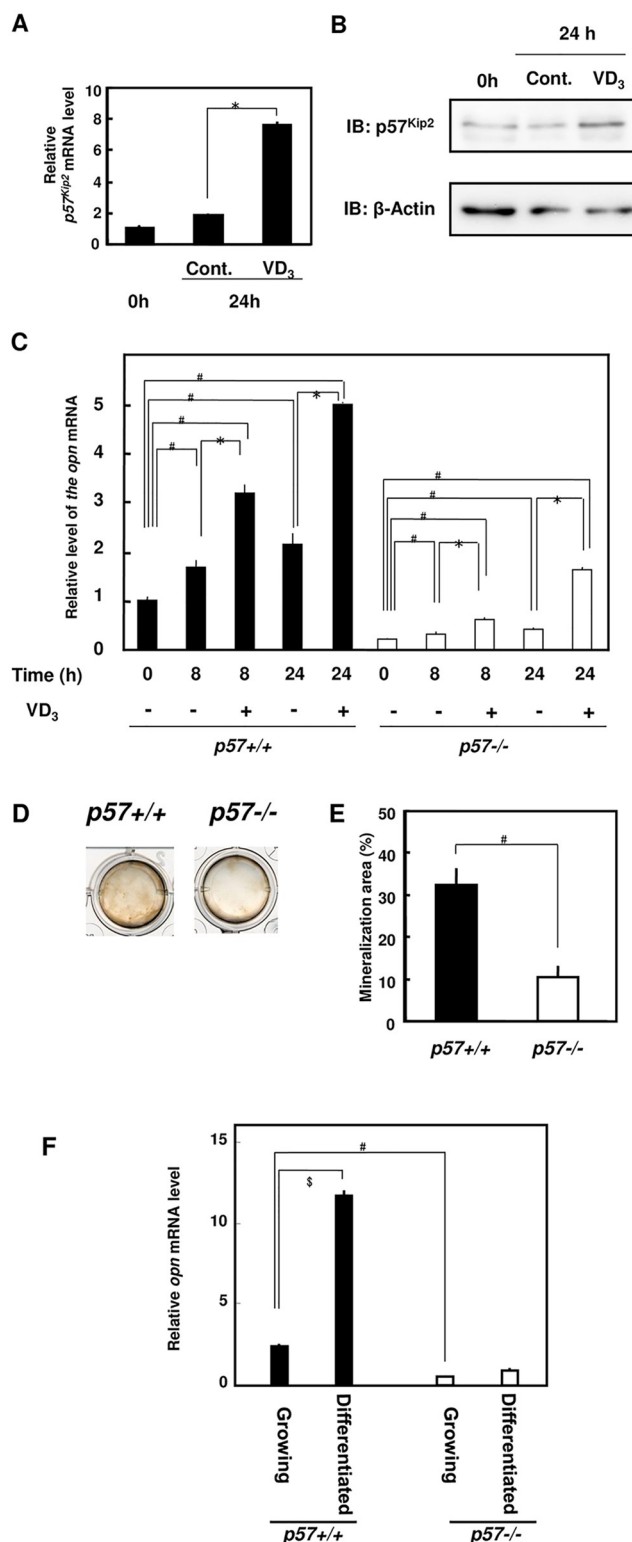

**Fig 3. Effects of the ablation of p57^Kip2 on primary cultured osteoblasts.** (A) Results of the quantitative RT-PCR analysis of *p57^Kip2* transcripts. The results were normalized to *hprt* levels. An asterisk indicates a statistically significant difference: *, $p < 0.01$. (B) Immunoblot showing the levels of p57^Kip2. In A and B, cells were treated with the vehicle (0.1% ethanol: Cont.) or 10 nM 1,25-(OH)$_2$VD$_3$ (VD3) for 24 h. (C) The effect of 10 nM 1,25-(OH)$_2$VD$_3$ on the expression of the *osteopontin* mRNA was analyzed using quantitative RT-PCR. The results were normalized to the *hprt*

mRNA. Under equal conditions, an asterisk indicates a statistically significant difference between 0.1% ethanol and 1,25-(OH)$_2$VD$_3$: *, $p < 0.01$. The # symbol indicates a statistically significant difference between *p57+/+* and *p57-/-* cells: #, $p < 0.01$. (D) Images of von Kossa staining of nodule mineralization in extended-culture osteoblastic cells. Primary osteoblasts were plated in 12-well multiplates, grown to confluence and incubated for 21 days with ascorbic acid, β-glycerophosphate and dexamethasone. (D) Photo of von Kossa-stained cells. (E) Graph of the analysis of the cells shown in (D) [mineralization area (%)]. An asterisk indicates a statistically significant difference: *, $p < 0.01$. (F) The expression of the *osteopontin* mRNA in *p57+/+* and *p57-/-* osteoblasts was analyzed using quantitative RT-PCR. Cells grown to 70% confluence were "growing cells", and mineralized confluent cells were "differentiated cells". The results were normalized to the *hprt* mRNA. The # symbol indicates a statistically significant difference between *p57+/+* and *p57-/-* cells: #, $p < 0.01$. The $ symbol indicates a statistically significant difference between growing and differentiated cells: $, $p < 0.01$.

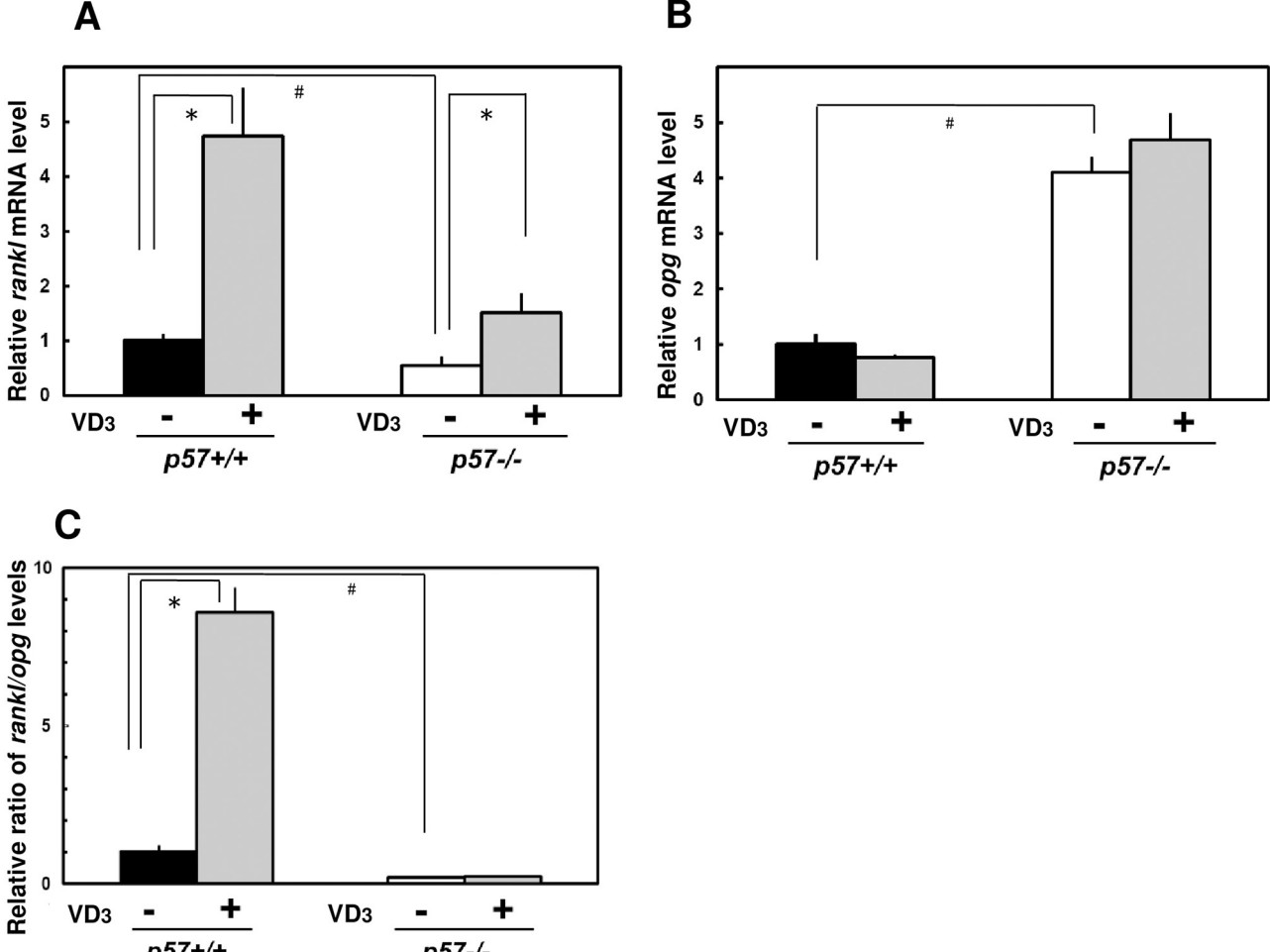

**Fig 4. VD$_3$-induced *rankl* expression in osteoblasts depended on p57$^{Kip2}$.** (A) Quantitative RT-PCR analysis of the levels of the *rankl* mRNA in osteoblasts treated with 0.1% ethanol (-) or 10 nM 1,25-(OH)$_2$VD$_3$ (+) for 72 h. Levels of the *rankl* mRNA were normalized to the mRNA levels of the constitutive gene *hprt1*. (B) Quantitative RT-PCR analysis of the expression of the *opg* mRNA in osteoblasts treated with 0.1% ethanol (-) or 10 nM 1,25-(OH)$_2$VD$_3$ (+) for 72 h. Levels of the *opg* mRNA were normalized to the mRNA levels of the constitutive gene *hprt1*. (C) The relative ratio of the *rankl* mRNA/*opg* mRNA was calculated and compared between *p57+/+* and *p57-/-* osteoblasts. In the graphs, relative levels in *p57+/+* cells were estimated. The # symbol indicates a statistically significant difference between *p57+/+* and *p57-/-* cells: #, $p < 0.01$. The asterisk indicates a statistically significant difference between 0.1% ethanol (-) and 10 nM 1,25-(OH)$_2$VD$_3$ (+): *, $p < 0.01$.

cells (Fig 4A). The expression of *rankl* is regulated by VD$_3$-VDR activation [30–32]. The p57$^{Kip2}$ deficiency reduced cellular responses to 1,25-(OH)$_2$VD$_3$. In contrast to *rankl*, *opg* expression was not altered by 1,25-(OH)$_2$VD$_3$ in both *p57+/+* and *p57-/-* cells (Fig 4B). The ratio of *rankl*/*opg* expression may be useful as an indicator of the osteoclastogenic activity of osteoblasts [32, 33]. As shown in Fig 4D, *p57+/+* cells displayed higher *rankl*/*opg* ratios than *p57-/-* cells. Treatment with 1,25-(OH)$_2$VD$_3$ significantly increased the ratio in *p57+/+* cells (Fig 4C). From these results, we concluded that the ablation of p57$^{Kip2}$ upregulated *opg*, suppressed 1,25-(OH)$_2$VD$_3$-dependent *rankl* expression, and resulted in defects in osteoclastogenic activities in osteoblasts.

## Discussion

As shown in the current study, p57$^{Kip2}$ specifically interacted with VDR and was required for osteoclastogenesis activities in osteoblasts. Studies using knockout mice indicated that p57$^{Kip2}$ functions mainly as a CDKI in mouse osteoblasts, because the lack of genes encoding other Cip/Kip family molecules, including both p21$^{Cip1}$ and p27$^{Kip1}$, did not result in abnormal bone formation, which was observed in *p57-/-* mice [11–13].

The degradation of p57$^{Kip2}$ via the proteasome pathway inhibits osteoblast maturation [34–37]. Thus, the expression of p57$^{Kip2}$ is necessary for osteoblast maturation. Following the evaluation of a novel ubiquitin ligase, FBL12, which is involved in TGF-1β-induced degradation of p57$^{Kip2}$, Kim et al. [36] noted that p57$^{Kip2}$ overexpression promotes the differentiation of primary osteoblasts. VD$_3$ increased p57$^{Kip2}$ levels in mouse osteoblasts. We revealed the VD$_3$-dependent upregulation of p57$^{Kip2}$ in mouse osteoblasts, and an increase in levels of the p57$^{Kip2}$ protein might also be associated with an increase its stabilization.

In our study, VDR was specifically associated with the CDKI domain of p57$^{Kip2}$. Cip/Kip molecules contain a characteristic CDKI domain (Fig 5). The p57$^{Kip2}$ CDKI domain contains the specific hydrophilic AELNAEDQN peptide and hydrophobic PLRGPGRLQ peptide. The 3D structure of p57$^{Kip2}$ has not yet been reported, but these p57$^{Kip2}$-specific peptides might be involved in the interaction with VDR. As shown in the study by Valcheva et al. [38], G0-synchronized primary VDR-deficient vascular smooth muscle cells express p57$^{Kip2}$ at higher levels than wild-type cells. These findings prompted us to investigate the regulation of p57$^{Kip2}$ levels by the VDR complex. The LBD of VDR was responsible for the interaction with p57$^{Kip2}$. This peptide of VDR contains the common LBD for other nuclear receptors, including PXR, LXR, FXR THR and RXR. According to our results, p57$^{Kip2}$ might interact with various nuclear receptors to regulate their activity. Joseph B et al. reported that p57Kip2 cooperated with Nurr1, the same nuclear receptor as VDR, and activated transcriptional activity at its transcription factor binding site (NBRE). Therefore, we investigated whether p57Kip2 also cooperated with VDR to activate transcriptional activity at the transcription factor binding site (VDRE) [39]. The coexistence of p57 and VDR facilitated 1,25-(OH)$_2$VD$_3$-dependent VDR activation, and p57 functioned as a cofactor of VDR.

```
AAH05412.1 21-67  (P57) [Mus musculus]           1:SLFGPVDHEELGRELRMRLAELNAE-DQNR---WDENFQQDVPLRGPGRLQ
AAH67842.1 32-78  (p57, Kip2) [Homo sapiens]     1:SLFGPVDHEELSRELQARLAELNAE-DQNR---WDYDFQQDMPLRGPGRLQ
AAH01971.1 31-75  (p27, Kip1) [Homo sapiens]     1:NLFGPVDHEELTRD---LEK-HCRDMEEASQRKWNFDFQNHKPLEGKYE--
AAH13967.1 21-63  (p21, Cip1) [Homo sapiens]     1:-LFGPVDSEQLRRDCDALMA-GCIQ--EARER-WNFDFVTETPLEGDF---
```

**Fig 5. CDKI domains of Cip/Kip family molecules.** Characteristic CDKI domains were compared among mouse (*Mus musculus*) p57$^{Kip2}$, human (*Homo sapiens*) p57$^{Kip2}$, p27$^{Kip1}$ and p21$^{Cip1}$. Identical amino acid residues in 3 of 4 peptides are enclosed with a red line. A blue line surrounds p57$^{Kip2}$-specific peptides, which were shared by mouse and human sequences.

As shown in Fig 4, p57<sup>Kip2</sup> regulated *opg* expression in mouse osteoblasts, in contrast to *opn* and *rankl*. The osteoclastogenic activities of osteoblasts depend on Rankl and Opg. Rankl is a membrane-bound signal transducer responsible for the differentiation and maintenance of osteoclasts; in addition, Rankl promotes osteoclast differentiation. In conjunction with the differences in *rankl* mRNA levels, *p57-/-* osteoblasts were inferior to *p57+/+* cells in terms of osteoclast induction activity. As shown in our current study, p57<sup>Kip2</sup> regulated *opg* expression, and the ablation of p57<sup>Kip2</sup> significantly upregulated the expression of the *opg* mRNA. Opg is the decoy receptor for Rankl to prevent osteoclastogenesis. Defects in osteoclastogenesis due to a lack of p57<sup>Kip2</sup> might result from both an increase in *opg* expression and the downregulation of *rankl*. Several studies have described the regulation of *opg* expression in osteoclastogenesis [20, 27, 40–42], but no information is available on *opg* downregulation. When opg was discovered by Yasuda et al. [41], most researchers believed that opg expression was vitamin D-dependent. However, Nakamichi et al. [43] reported that rankl mRNA expression was VDR dependent but opg expression in osteoblast-specific KO mice did not change after treatment with 1, 25 vitamin D3. These results suggested that opg expression might not be VDR dependent. Recently, cancer cells were shown to release more Opg than normal cells [44–48]. Because p57<sup>Kip2</sup> is a tumor suppressor protein, p57<sup>Kip2</sup>-deficient cancer cells may express and release Opg.

The levels of the *opn* mRNA in osteoblasts are consistent with the *in vivo* osteomalacia-like phenotype; however, *in vitro*, *p57-/-* osteoblasts cultured with maturation medium produced fewer mineralized nodules than *p57+/+* cells. The expression of the *opn* mRNA was also upregulated by $1,25\text{-}(OH)_2VD_3$-VDR activity, similar to *rankl*. Kitazawa et al. [30–32] previously reported that $1,25\text{-}(OH)_2VD_3$ enhances osteoclastogenesis via the transactivation of the *rankl* gene in osteoblasts through the VDRE in both humans and mice. Urano et al. [29] observed the upregulation of *p57<sup>Kip2</sup>* transcripts and increased levels of the p57<sup>Kip2</sup> protein in rat osteoblasts (*p57+/+*) cultured in the presence of $1,25\text{-}(OH)_2VD_3$. We expected that sufficient VDR activation by $1,25\text{-}(OH)_2VD_3$ activity might be mediated by p57<sup>Kip2</sup> during osteoblast maturation. An assessment of the SaOS2 tet-off p57<sup>Kip2</sup> stable cell line yielded data consistent with our expectations, at least in terms of the activation of the *opn* promoter containing the VDRE. Of course, our results might be partially attributed to the interaction of p57<sup>Kip2</sup> and VDR in osteoblasts and might depend on the other pathway via CDK activation.

In the present study p57<sup>Kip2</sup> formed complexes with VDR and function as a cell cycle regulator and a mediator of the $1,25\text{-}(OH)_2VD_3$-induced transcriptional activation of osteoblast genes in mineralizing osteoblastic cells. These data identified possible roles for p57<sup>Kip2</sup> in regulating osteoblast differentiation and bone metabolism.

## Supporting information

**S1 Raw images.**
(PDF)

**S1 File.**
(PPTX)

## Acknowledgments

We would like to thank T Usui for providing the SaOS2 tet-off p57<sup>Kip2</sup> cells. We would also like to thank T Tomuro, A Nara, Y Takahashi, R Oikawa, Dr. A Karakawa, Dr. Y Nakamichi, Prof. S Inoue, Prof. H Itabe and Prof. M Tomita for their support of this study. In addition, we wish to thank Prof. E Abe for critically reading the manuscript.

## Author Contributions

**Conceptualization:** Katsuhiko Takahashi, Hitoshi Amano, Tomohiko Urano.

**Data curation:** Katsuhiko Takahashi, Hitoshi Amano, Tomohiko Urano, Minqi Li.

**Formal analysis:** Katsuhiko Takahashi, Hitoshi Amano, Tomohiko Urano.

**Funding acquisition:** Hitoshi Amano, Tomohiko Urano, Nobuaki Higashi.

**Investigation:** Hitoshi Amano.

**Methodology:** Katsuhiko Takahashi, Hitoshi Amano.

**Project administration:** Katsuhiko Takahashi, Hitoshi Amano.

**Resources:** Hitoshi Amano.

**Supervision:** Hitoshi Amano, Kazuhiro Aoki, Norio Amizuka, Keiichi I. Nakayama, Keiko Nakayama, Nobuyuki Udagawa, Nobuaki Higashi.

**Validation:** Hitoshi Amano.

**Visualization:** Minqi Li, Norio Amizuka.

**Writing – original draft:** Katsuhiko Takahashi, Hitoshi Amano, Tomohiko Urano.

**Writing – review & editing:** Hitoshi Amano, Meiko Oki.

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
