## [Decision Letter · Decision Letter 0]

23 Feb 2022

PONE-D-21-23783p57Kip2 is an essential regulator of vitamin D receptor-dependent mechanismsPLOS ONE

Dear Dr. Amano,

Thank you for submitting your manuscript to PLOS ONE. After careful consideration, we feel that it has merit but does not fully meet PLOS ONE’s publication criteria as it currently stands. Therefore, we invite you to submit a revised version of the manuscript that addresses the points raised during the review process. Please submit your revised manuscript by Apr 09 2022 11:59PM. If you will need more time than this to complete your revisions, please reply to this message or contact the journal office at plosone@plos.org. Please include the following items when submitting your revised manuscript:A rebuttal letter that responds to each point raised by the academic editor and reviewer(s). You should upload this letter as a separate file labeled 'Response to Reviewers'.A marked-up copy of your manuscript that highlights changes made to the original version. You should upload this as a separate file labeled 'Revised Manuscript with Track Changes'.An unmarked version of your revised paper without tracked changes. You should upload this as a separate file labeled 'Manuscript'.

We look forward to receiving your revised manuscript.

Kind regards,

Chandi C. Mandal, Ph.D.

Academic Editor

PLOS ONE

Journal Requirements:

Additional Editor Comments (if provided):

Your manuscript has been carefully evaluated by subject experts. It needs a major revision. See the comments below.

Reviewers' comments:

Reviewer's Responses to Questions

5. Review Comments to the Author

Reviewer #1: The Katsuhiko Takahashi et al., addresses or responded both the reviewers comments and potentially improved the revised manuscript.

The authors were reworked and improved the tiltle of the article along with the typo errors which was metioned earlier was rectified. Hence I recommend the article to be published

Reviewer #2: Overall comments:

The format of the manuscript is good. The author is applauded for the overall cover image.

As this study is based on a hypothesis, reasoning to every statement made is important.

The author has failed to provide references and reasons in the paper's main section i.e., the author can add references to the procedure used and why this particular procedure was chosen. Elaboration to the contents of result section is to be given, reasoning of certain statements given by the author is expected or if taken from different articles references to the same is expected.

INTRODUCTION

"Transfection of p57Kip2 into Saos-2 osteosarcoma cells induced arrest at G1 phase through a mechanism that does not appear to require Rb or p53."- The author can give a short reasoning for the mentioned sentence.

HISTOLOGY AND IMMUNOHISTOCHEMISTRY

Why was this procedure considered? Any particular references?

NODULE MINERALIZATION

"Following an additional 3 weeks of culture.."- Why three weeks of culture preferred? Reference for this particular statement to be added by the author.

QUANTITATIVE RT-PCR

"The relative levels of the mouse p57Kip2 and ......, were determined using the comparative Ct (cycles at threshold fluorescence) method."- Why was this method chosen, reasoning to be given by the author.

"The relative levels of the human p57Kip2...........reference gene glyceraldehyde-3-phosphate dehydrogenase (gapdh), were also determined."- was it by the same approach or a different. What are the different approaches that can be used here to determine the relative levels?

"The sequences of the PCR primers........"- How were the forward and reverse primer selected and how was it obtained?

p57Kip2 245 enhances the transcriptional activities of VDR

"We performed a luciferase assay employing a reporter plasmid with the opn 5' flanking region-luciferase cDNA to investigate the role of p57Kip2 254 in the 255 activation of the VDRE."- Why was this investigation done particularly?

p57-/- osteoblasts exhibit defects in osteoclastogenesis

"Opg, also known as an osteoclastogenesis inhibitory factor, functions as a decoy receptor for Rank to obstruct Rankl-Rank signaling and inhibit osteoclastogenesis."- Reference?

"Rankl expression might depend on p57Kip2 347 to some extent, and opg expression might be suppressed by p57Kip2." Why does it depend on p57Kip2?

---

## [Author Response · Author response to Decision Letter 0]

16 Jun 2022

Review Comments to the Author

Reviewer #1: The Katsuhiko Takahashi et al., addresses or responded both the reviewers comments and potentially improved the revised manuscript.

Response

Thank you for your helpful comments on the manuscript. It has been substantially improved. We appreciate your efforts.

Reviewer #2: Overall comments:

The format of the manuscript is good. The author is applauded for the overall cover image.

As this study is based on a hypothesis, reasoning to every statement made is important.

The author has failed to provide references and reasons in the paper's main section i.e., the author can add references to the procedure used and why this particular procedure was chosen. Elaboration to the contents of result section is to be given, reasoning of certain statements given by the author is expected or if taken from different articles references to the same is expected.

INTRODUCTION

"Transfection of p57Kip2 into Saos-2 osteosarcoma cells induced arrest at G1 phase through a mechanism that does not appear to require Rb or p53."- The author can give a short reasoning for the mentioned sentence.

Response Thank you for your helpful comment. Since several reports showed that SAOS2 cells were defective for p53 and Rb [Hinds et al. (1992), van der Heudel and Harlow (1993)], transfection of p57Kip2 into Saos-2 osteosarcoma cells induced arrest at G1 phase through a mechanism that does not appear to require Rb or p53 [Matsuoka et al., (1995)].

Reference:　 Hinds, P.W., S. Mittnacht, V. Dulic, A. Arnold, S.I. Reed, and R.A. Weinberg. Regulation of retinoblastoma protein functions by ectopic expression of human cyclins. Cell　1992; 70: p993-1006.

van der Heuvel, S. and E. Harlow. Distinct roles for cyclin dependent kinases in cell cycle control. Science,1993; 26: p2050-2054.

HISTOLOGY AND IMMUNOHISTOCHEMISTRY

Why was this procedure considered? Any particular references?

Response It is a description for S Figure 2.

NODULE MINERALIZATION

"Following an additional 3 weeks of culture."- Why three weeks of culture preferred? Reference for this particular statement to be added by the author.

Response We followed the methods of the reference papers below.

Marzia M, Sims NA, Voit S, Migliaccio S, Taranta A, Bernardini S, Faraggiana T, Yoneda T, Mundy GR, Boyce BF, Baron R, and Teti A. Decreased c-Src expression enhances osteoblast differentiation and bone formation. J Cell Biol.2000; 151: p311-320.

QUANTITATIVE RT-PCR

"The relative levels of the mouse p57Kip2 and ......, were determined using the comparative Ct (cycles at threshold fluorescence) method."- Why was this method chosen, reasoning to be given by the author.

"The relative levels of the human p57Kip2...........reference gene glyceraldehyde-3-phosphate dehydrogenase (gapdh), were also determined."- was it by the same approach or a different. What are the different approaches that can be used here to determine the relative levels?

"The sequences of the PCR primers........"- How were the forward and reverse primer selected and how was it obtained?

Response We must sincerely apologize. The description of M & M is incorrect. We determined the relative levels of human and mouse p57Kip2 in the same way. (Reference,　T Urano, M Shiraki, H Yagi, M Ito, N Sasaki, M Sato, Y Ouchi, and S Inoue. GPR98/Gpr98 gene is involved in the regulation of human and mouse bone mineral density. J Clin Endocrinol Metab 2012; 97: E565-574). We have revised this section and added this reference.

p57Kip2 245 enhances the transcriptional activities of VDR

"We performed a luciferase assay employing a reporter plasmid with the opn 5' flanking region-luciferase cDNA to investigate the role of p57Kip2 254 in the 255 activation of the VDRE."- Why was this investigation done particularly?

Response Joseph B et al. reported that p57Kip2 cooperated with Nurr1, the same nuclear receptor as VDR, and activated transcriptional activity at its transcription factor-binding site (NBRE). Therefore, we investigated whether p57Kip2 also cooperated with VDR to activate transcriptional activity at the transcription factor-binding site (VDRE). We added the above sentences to explain our investigation in the discussion. Reference:　Joseph B, Wallén-Mackenzie A, Benoit G, Murata T, Joodmardi E, Okret S, and Perlmann T. 　p57(Kip2) cooperates with Nurr1 in developing dopamine cells. Proc Natl Acad Sci U S A. 2003; 100: p15619-15624.

p57-/- osteoblasts exhibit defects in osteoclastogenesis　　　　　　　　　　　　　　　　　　　　　　　　　　　　　　　　　　　　"Opg, also known as an osteoclastogenesis inhibitory factor, functions as a decoy receptor for Rank to obstruct Rankl-Rank signaling and inhibit osteoclastogenesis."- Reference?　　　　　　　　　　　　　　　　　　　　　　　　　　　　　　　　Response Please refer to Reference 36. 　　 [36] Yasuda H, Shima N, Nakagawa N, Mochizuki SI, Yano K, Fujise N, Sato Y, Goto M, Yamaguchi K, Kuriyama M, Kanno T, Murakami A, Tsuda E, Morinaga T, Higashio K. Identity of osteoclastogenesis inhibitory factor (OCIF) and osteoprotegerin (OPG): a mechanism by which OPG/OCIF inhibits osteoclastogenesis in vitro. Endocrinology 1998;139:1329-37.

　

"Rankl expression might depend on p57Kip2 347 to some extent, and opg expression might be suppressed by p57Kip2." Why does it depend on p57Kip2?　　　　　　　　　　　　　　　　　　　　　　　　　　　　　　　　　　　　　　　　　　　　　　　　　　　　　　　　　　　　　　　　 　Response Thank you for your helpful suggestions. We believe that p57Kip2 might regulate the expression of VDR-dependent genes by associating with VDR. When opg was discovered by Yasuda et al., most researchers believed that opg expression was vitamin D dependent. However, Nakamichi et al. reported that rankl mRNA expression was VDR dependent, but opg expression in osteoblast-specific KO mice did not change after treatment with 1, 25 vitamin D3. These results suggested that opg expression might not be VDR dependent. We observed the expression of opg mRNA in proliferative osteoblasts deficient in the CDKI gene.

References: Nakamichi Y, Udagawa N, Horibe K, Mizoguchi T, Yamamoto Y, Nakamura T, Hosoya A, Kato S, Suda T, Takahashi N. VDR in Osteoblast-Lineage Cells Primarily Mediates Vitamin D Treatment-Induced Increase in Bone Mass by Suppressing Bone Resorption. J Bone Miner Res. 2017 Jun;32(6):1297-1308.

---

## [Decision Letter · Decision Letter 1]

17 Oct 2022

p57Kip2 is an essential regulator of vitamin D receptor-dependent mechanisms

PONE-D-21-23783R1

Dear Dr. Amano,

We’re pleased to inform you that your manuscript has been judged scientifically suitable for publication and will be formally accepted for publication once it meets all outstanding technical requirements.

Kind regards,

Chandi C. Mandal, Ph.D.

Academic Editor

PLOS ONE

Additional Editor Comments (optional):

Revision has improved quality of the manuscript which may be accepted for its publication.

---

## [Editor Report · Acceptance letter]

20 Oct 2022

PONE-D-21-23783R1 

p57^Kip2^ is an essential regulator of vitamin D receptor-dependent mechanisms 

Dear Dr. Amano:

I'm pleased to inform you that your manuscript has been deemed suitable for publication in PLOS ONE. Congratulations! Your manuscript is now with our production department. 

Kind regards, 

on behalf of

Dr. Chandi C. Mandal 

Academic Editor

PLOS ONE